# Digital Educational Support Groups Administered through WhatsApp Messenger Improve Health-Related Knowledge and Health Behaviors of New Adolescent Mothers in the Dominican Republic: A Multi-Method Study

**Samantha Stonbraker** [1,2,*] **, Elizabeth Haight** [3] **, Alana Lopez** [3] **, Linda Guijosa** [3] **, Eliza Davison** [3] **, Diane Bushley** [3] **, Kari Aquino Peguero** [2] **, Vivian Araujo** [2] **, Luz Messina** [2] **and Mina Halpern** [2]

1    College of Nursing, University of Colorado Anschutz Medical Campus, Aurora, CO 80045, USA
2    Clínica de Familia La Romana, La Romana 22000, Dominican Republic;
     DraKarinaAquino@gmail.com (K.A.P.); vivianaraujocf@gmail.com (V.A.); LuzMessinacf@gmail.com (L.M.);
     Mina@ClinicadeFamilia.org.do (M.H.)
3    Planned Parenthood of the Great Northwest and the Hawaiian Islands, Seattle, WA 98122, USA;
     Liz.Haight@ppgnhi.org (E.H.); Alana.Lopez@ppgnhi.org (A.L.); Linda.Guijosa@ppgnhi.org (L.G.);
     Eliza.Davison@ppgnhi.org (E.D.); Diane.Bushley@ppgnhi.org (D.B.)
*    Correspondence: Samantha.Stonbraker@CUAnschutz.edu

**Abstract:** (1) *Background*: In limited-resource settings such as the Dominican Republic, many factors contribute to poor health outcomes experienced by adolescent mothers, including insufficient support and/or health knowledge. In response, we designed a digital educational support group, administered through WhatsApp Messenger, for new adolescent mothers. The purpose of this study was to assess if participation in this digital support group could improve health outcomes and health behaviors. (2) *Methods*: Participants completed questionnaires with a health literacy screener, demographic items, knowledge questions, the Index of Autonomous Functioning, and five Patient Reported Outcomes Measurement Information System scales before and after the moderator-led intervention. Differences between pre- and post-intervention scores were calculated and perceptions of the intervention were explored through in-depth interviews analyzed with content analysis. Participants' well-baby visit attendance and contraceptive use were compared to that of controls and a national sample. (3) *Results*: Participants' ($N$ = 58) knowledge scores increased ($p < 0.05$). Participants were 6.58 times more likely to attend well-baby visits than controls (95% CI: 2.23–19.4) and their contraceptive use was higher than that of the national sample ($p < 0.05$). Participants indicated the intervention was enjoyable and beneficial. (4) *Conclusion*: This adolescent-centered digital intervention is a promising method to improve health outcomes and health behaviors of young mothers in limited-resource settings.

**Keywords:** digital health; mHealth; adolescent mother; digital support group; nursing informatics; WhatsApp Messenger

## 1. Introduction

The Dominican Republic (DR) has one of the highest rates of adolescent pregnancy in the world, as over 20% of girls aged 15–19 years are, or have been, pregnant [1,2]. This is concerning, as adolescent pregnancy may lead to negative health outcomes for the mother and baby [3–7]. Additionally, adolescent pregnancy can alter girls' life courses, by limiting their educational attainment and decreasing their lifetime earning potential, which has implications for broader society as well [8–10]. The postpartum

period and transition into motherhood is a stressful time for any mother, and may be particularly difficult for adolescents to manage, but how to effectively support girls living in limited-resource settings during this time is insufficiently researched and few tools are available to provide effective guidance [11,12].

The successful management of any mother's and baby's health in the postpartum period requires new mothers to acquire much information regarding how to take care of themselves and their newborn [13,14]. Topics such as when and how to feed their baby and how to manage their own emotional and physical needs are just a few of the many information needs of new mothers [14]. Obtaining this information may be especially difficult for adolescents in limited-resource settings, as they undergo the same physical and psychological changes during pregnancy and childbirth but may lack the resources and life experience their older counterparts have to help manage them [12,15]. In addition to information needs, adolescents in the postpartum period require support, both from professional and social sources [16–19]. While professional support can offer pertinent health care services, social support is necessary for effective self- and infant care and has been shown to reduce postpartum depression and improve other maternal health outcomes [18,19].

Because mobile health (mHealth) technologies are rapidly advancing and are ubiquitous even in limited-resource settings, there is much potential for digital solutions to extend high quality health services to hard-to-reach populations [20–24]. Therefore, to help simultaneously meet the health information and various support needs of new adolescent mothers living in a limited-resource setting, La Romana in the DR, during the postpartum period, our group developed a digital educational support group called Fortaleciendo la Autodeterminación de Madres Adolescentes (FAMA) or "Strengthening the Self-determination of Adolescent Mothers" [25]. This group was designed for WhatsApp Messenger (hereafter referred to as WhatsApp), one of the most widespread social media applications (app) in the world, which is increasingly being used to promote health [26–29]. Of note, WhatsApp contains end-to-end encryption, which increases the level of privacy for end-users [26].

A known benefit of digital health platforms is that patients can use them to connect to their providers, as well as to their peers [30,31]. There is also evidence that professional and social support can be obtained, at least partially, through electronic communication [17,32,33]. The FAMA intervention takes advantages of these benefits and is particularly relevant to adolescents, as cellular phones are one of the most common ways this age group communicates [34,35]. Additionally, previous studies have shown that adolescents are willing to report and discuss health through text messaging, especially when their privacy is guaranteed [36]. Previous studies on support groups run through WhatsApp, including one for new mothers, demonstrate that these groups are safe and feasible [27,37]. Notably, many studies have looked at interventions and ways to improve health education on a variety of topics with adolescents, but few have reported on outcomes [23,38,39].

Therefore, the purpose of this study was to assess if participation in our digital educational support group, administered through WhatsApp Messenger through a local health clinic, could improve the following health outcomes among new adolescent mothers: postpartum-related health knowledge, use of modern contraception, attendance at scheduled well-baby appointments, levels of perceived support, and autonomous functioning.

## 2. Materials and Methods

### 2.1. Study Design

To enable a comprehensive evaluation of the FAMA intervention, this multi-method study had three data collection components.

1.　Baseline and follow-up surveys. Intervention participants completed surveys that assessed the outcome measures of interest before and after participation in the intervention.
2.　Medical record review. Clinical data (attendance at well-baby visits, modern contraceptive use, and completion of scheduled vaccinations) were extracted from FAMA participants' medical

records and compared to similar data extracted from the medical records of a control group. Where control group data were insufficient, participants' medical record data were compared to data from a publicly available national demographic and health survey.

3.  In-depth interviews. Participants who completed the FAMA intervention and attended the closing group session were invited to complete an in-depth interview to assess their perspectives of the FAMA intervention and identify areas for future improvement.

### 2.2. Ethical Approval

This study was approved by two institutional review boards in the DR—the Instituto Dermatológico y Cirugía de Piel "Dr. Huberto Bogaert Díaz" and the Consejo Nacional de Bioética en Salud.

### 2.3. Research Setting

The digital educational support groups were designed and led through the Módulo de Adolescentes Materno Infantil (MAMI), a clinic that offers health care services and education to approximately 3000 adolescents in La Romana, DR. At MAMI, family planning counseling and methods are offered, as is sexual and reproductive health education, pre- and post-natal care, pediatric care, and testing for sexually transmitted infections.

### 2.4. Study Team

MAMI has closely collaborated with Planned Parenthood of the Great Northwest and the Hawaiian Islands (PPGNHI) for nearly a decade. As part of this long-standing partnership, staff from PPGNHI's Global Programs and Education departments regularly travel to the DR to provide support for MAMI's health education programs. Our study team, therefore, consisted of bilingual PPGNHI staff, MAMI educators and administrators, as well as the research department and administrators at Clínica de Familia La Romana, the parent organization that supports MAMI.

### 2.5. Participants and Recruitment

The target sample size agreed upon was 60 adolescents, as determined by funding and logistical considerations. Some participants (*n* = 16) had participated in initial user-centered design sessions and had agreed to participate in this study phase. Therefore, the study team recruited 44 additional adolescent mothers in the waiting room at MAMI. Inclusion criteria were that FAMA participants be 14 to 19 years of age, have at least one child, have given birth in the previous six months, speak Spanish, live in La Romana, and attend pediatric appointments at MAMI. The babies of participants who had participated in the first phase were older (up to 11 months) at the time the intervention began. Exclusion criteria were not having the capacity to understand the study or provide informed consent. As Dominican law 136-03 classifies adolescents who are pregnant and/or have had children as emancipated minors who can consent to participate in research and/or health care services, those who agreed to participate provided written informed consent after study team members thoroughly explained the study and answered any questions.

### 2.6. FAMA Intervention

#### 2.6.1. Development

The methods used to establish the content for the FAMA intervention are fully described elsewhere [25]. However, to provide a general description, a group of target participants (*n* = 24) completed participatory user-centered design sessions that explored the health information needs of new adolescent mothers in the DR as well as their experiences as new mothers. Participants first identified information and support needs, then worked with each other to prioritize the health information and support topics that were the most meaningful and potentially important to include

in the intervention. Growth and development stages of infants, understanding your baby, common infant illnesses, breastfeeding, nutrition during a child's first year, and family planning were the topics that participants agreed were the most important to address. We then identified pertinent content associated with each identified topic through a brief literature review and established the guide for the FAMA intervention with the corresponding findings.

### 2.6.2. Pre-Intervention

We divided FAMA participants into three separate intervention groups ($n = 16$, $n = 20$, $n = 22$), to enable meaningful participation for each participant in each group. All groups began with an in-person meeting where participants met the intervention moderators (a health educator and physician/study coordinator at MAMI), received a cellular phone, instructions for how to access cellular data packages (replenished weekly), met other participants in the group, discussed intervention procedures, established group norms and guidelines, and completed baseline surveys. If a participant from the first group missed the in-person meeting for their group, they were able to move to the later groups, which accounts for the larger group size in the second and third groups. Two participants did not attend any in-person group sessions and were removed from the study for a final $N = 58$.

### 2.6.3. Intervention

Each group participated in the intervention for 12 weeks to allow sufficient time for participants to adapt to the intervention format, learn about and ask questions pertaining to each topic, and to enable observable changes in outcome measures. During the intervention, informational messages and associated images from the previously developed manual [25] were sent to the WhatsApp groups on weekdays. Intervention moderators interacted with participants by providing support and answering questions in the group chat. Moderators encouraged participants to respond to the informational messages with questions, comments, and their own experiences. To confirm message receipt and ongoing participation, participants were asked to at least respond with a thematic "emoji of the day", which was included in each message. There were also biweekly optional quizzes and a biweekly forum with a question posed to the group to reflect upon.

### 2.6.4. Post-Intervention

A second in-person group was held after the intervention was complete, in which participants completed follow-up surveys and a final reflection activity, and a convenience sample of participants completed in-depth interviews to gather participants' perspectives on the intervention, including suggestions for improvement.

### 2.7. Measures and Data Analysis: Baseline and Follow-Up Surveys

### 2.7.1. Demographics and Postpartum/Neonatal Health Knowledge

At enrollment, a demographic survey was administered that included questions regarding participants' age, marital status, level of education, employment status, and number of children. Health literacy was measured at baseline with a single self-reported health literacy screening question that has been previously validated in Spanish [40,41]. Participants answered the question, "how confident are you filling out medical forms by yourself?" with one of the responses: (a) not at all confident, (b) a little confident, (c) somewhat confident, (d) very confident, or (e) completely confident. Those who responded "a" or "b" were classified as likely to have inadequate health literacy and the rest were considered likely to have adequate health literacy [40,41]. Health knowledge was assessed with multiple-choice questions designed by the study team based on the health education topics to be provided in the intervention. There were 11 points possible, with scores calculated based on the number of correct responses.

### 2.7.2. Index of Autonomous Functioning

As autonomous functioning is associated with more engagement in learning, greater energy and vitality, lower stress, increased wellbeing, and more rewarding socialization [42–44], we included a validated assessment of autonomous functioning, the Index of Autonomous Functioning (IAF), to assess changes in this important outcome [42]. The IAF scale provides a measure of trait autonomy and is scored on three subscales [42]. We calculated a total composite score for participants who completed both a baseline and follow up survey using the average score of all 15 questions in the scale. Each five-question Likert subscale was scored; total possible scores ranged from 5 to 25 according to a Likert of 1 point (not true at all) to 5 points (completely true). Five questions were reverse scored. High scores on the authorship/self-congruence (when "a person is wholeheartedly willing to act on, or stand behind, what they do" [42,45]) subscale and the interest subscale indicate more autonomy and low averages of the susceptibility to control (when individuals behave in response to pressure from others [42]) subscale indicate more autonomy.

### 2.7.3. PROMIS Measures

To assess perceived levels of social support, five of the Patient Reported Outcomes Measurement Information System (PROMIS)® measures were administered: (1) emotional support, (2) informational support, (3) social isolation, (4) instrumental support, and (5) companionship. Benefits of using the PROMIS scores are that they have been extensively validated in diverse populations, including among Spanish speakers, they are freely available, and can be administered in a relatively short amount of time [46,47]. There were six questions in each of the PROMIS scales we administered. Response options for each question are scored on a 5-point Likert scale ranging from "never" (1 point) to "always" (5 points). Total possible points for each scale range from 6 to 30. Total scores from each scale were calculated for participants who completed both baseline and follow up assessments and then, as with all PROMIS measures, final scores were converted into a standardized T-score with a standard deviation of 10. Thus, a person with a T-score of 40 is one standard deviation below the arithmetic mean.

### 2.7.4. Data Analysis

Paired *t*-tests were used to assess differences in mean scores on all survey measures before and after the intervention.

### 2.8. Measures and Data Analysis: Medical Record Review

Data were extracted from FAMA participants' medical records as well as from the medical records of additional MAMI patients ($N$ = 22), "controls" who had similar characteristics to those of the FAMA participants (age of 14–19 years with a baby ≤6 months of age) and were enrolled in pediatric services during the intervention period. Data were collected from controls to enable comparisons between FAMA participants' and non-participants' attendance at well-baby visits and modern contraceptive use.

### 2.8.1. Attendance at Well-Baby Visits

The dates participants and controls attended their scheduled well-baby pediatric appointments during the 12-week intervention period and up to one month after were collected from medical records. At MAMI, the recommended well-baby visits are 15 days after birth and then monthly until one year of age. We categorized attendance at well-baby visits as good attendance (attended two or more recommended/scheduled visits) or poor attendance (attended one or fewer visits) during the four-month reporting period. We calculated an odds ratio to compare the well-baby visit attendance of participants and controls.

2.8.2. Modern Contraceptive Use

Participants' and controls' use of modern contraceptive prevalence (having any intrauterine device, oral contraceptive prescription, receiving the Depo Provera injection, or having an Implanon implant) was extracted from medical records. Whether or not a participant was using a modern method during the intervention period and up to one month after was dichotomized to yes or no. During data collection, we realized there were insufficient data on controls' family planning methods to enable meaningful comparisons. Therefore, we compared the modern contraceptive use of our study participants to that of a recent national sample of adolescents across the DR (24% of girls aged 15–19) using a Chi-square test [48].

2.8.3. Babies' Vaccination Status

Data regarding participants' and controls' babies' vaccination status were collected from copies of their World Health Organization and Ministry of Health endorsed vaccination cards kept in their medical records. However, due to a large percentage of missing vaccination data among both participants and controls, neither descriptive statistics nor case/control analyses could be performed for vaccination coverage.

*2.9. Measures and Data Analysis: In-Depth Interviews*

Post-intervention in-depth interviews were conducted by study team members using a semi-structured interview guide, developed by the study team, and were audio recorded. The guide included eight questions supported by several prompts that aimed to explore participants' perceptions of the FAMA intervention and recommendations for improvement. Interviewers all completed a training to ensure interviews were administered in the same format. Recordings were transcribed verbatim and verified by the study team. Transcripts were then analyzed through qualitative descriptive content analysis using Dedoose software [49–51]. Members of the study team generated a codebook through open coding of a selection of transcripts. Two study team members then independently coded transcripts with the initial codebook. Coder agreement was established by independently coding a selection of transcripts and double coding every ninth transcript. Discrepancies in coding and additional suggested codes were discussed until consensus was reached.

## 3. Results

*3.1. Demographics*

The median age of the 58 adolescent mothers who participated in the intervention was 18 (range 15–20) years at the time of enrollment (either for initial design sessions or for the intervention) (Table 1). Participants' babies had a median age of 3.6 months (IQR = 4 months) at the start of the intervention. Most participants (93%) were first time mothers. Forty percent of participants were enrolled in school, 97% were unemployed, 79% were married or had a domestic partner, and 67% were economically dependent on their partner.

**Table 1.** Demographic characteristics of participants (*N* = 58).

| Characteristic | *n* (%) |
|---|---|
| Health literate * | |
| Yes | 39 (69.6) |
| No | 17 (30.4) |
| Attends school | |
| Yes | 23 (40) |
| No | 35 (60) |
| Current grade in school or highest-grade level completed | |
| 6th Primaria (Middle School) | 2 (3.4) |
| 7th Primaria (Middle School) | 5 (8.6) |
| 8th Primaria (Middle School) | 9 (15.5) |
| 1st Bachiller (High School) | 16 (27.6) |
| 2nd Bachiller (High School) | 8 (13.8) |
| 3rd Bachiller (High School) | 13 (22.4) |
| 4th Bachiller (High School) | 5 (8.6) |
| Has a job | |
| Yes | 2 (3.4) |
| No | 56 (96.6) |
| On whom does the participant depend economically? | |
| Partner | 39 (67.2) |
| Parents | 7 (12.1) |
| Both Partner and Parents | 10 (17.2) |
| Other | 1 (1.7) |
| Missing | 1 (1.7) |
| Has a partner | |
| Yes | 46 (79.3) |
| No | 12 (20.7) |
| Number of children | |
| 1 | 54 (93.1) |
| 2 | 3 (5.2) |
| 3 | 1 (1.7) |
| | **Median, IQR** |
| Participant's age (years) | 18, 1 |
| Range of time with partner (months) | 24, 24 |
| Age of youngest child (months) * | 3.6, 4 |

Note: * 2 missing values for this question.

*3.2. Intervention*

Of participants, 44 (76%) completed the intervention (actively participated in groups through the final session, attended the last in-person meeting, and completed follow up surveys); 11/16 group 1 participants completed, 15/20 group 2 participants completed, and 18/22 group 3 participants completed. FAMA participants were very active with a total of 19,536 messages, 1,582 pictures, and 10,369 emojis sent in the three groups. The content of messages included asking questions about the daily and weekly health topics, sharing personal experiences, and telling stories about how their situation related to the current topic. Sent pictures were mainly of participants' newborns to provide context for questions about what was happening (i.e., a skin rash) or to share "cute" and/or meaningful moments. Moderators found themselves correcting several misconceptions about how to appropriately manage self- and infant care.

*3.3. Baseline and Follow-Up Surveys*

### 3.3.1. Health Knowledge

There was a statistically significant increase in health knowledge post-intervention as the mean percent of correct health knowledge questions increased from 74.3% to 86.4% between baseline and study end ($p < 0.05$) among the 44 participants who completed both baseline and post-intervention health knowledge questions.

### 3.3.2. Index of Autonomous Functioning

There was a slight decrease in the mean composite score on the IAF between the pre (46.1 SD ± 5.6) and the post (45.9 SD ± 5.6) questions among the 39 participants who completed both baseline and post-intervention IAF questions, though this change was not statistically significant.

### 3.3.3. PROMIS Measures

Participants who completed pre- and post-intervention PROMIS questions ($n = 39$) all had scores within one standard deviation of the population mean. The PROMIS scales Social Isolation, Instrumental Support, and Companionship increased in scores post-intervention, with an average increase of 1.3 points. The Emotional Support and Informational Support scales decreased between the pre- and post-intervention assessments, with an average decrease of 0.6 points. None of the changes in the PROMIS measures were statistically significant.

*3.4. Medical Record Review*

### 3.4.1. Attendance at Well-Baby Visits

Study participants with available medical records ($n = 57$) were 6.58 times more likely to attend at least two of their indicated well-baby visits than controls ($n = 22$) (95% CI: 2.23–19.4). This result is statistically significant at the 95% confidence interval.

### 3.4.2. Prevalence of Modern Contraceptive Use

Of the 56 participants with medical records with family planning data, 34 (61%) had used a modern contraceptive method between the birth of their youngest child and the start of the intervention and 41 (72%) either continued to use, or had started, a method during the intervention. Most participants used Depo Provera (58%) followed by an Implanon (9%), oral contraceptives (9%), and one participant had an intrauterine device (IUD). Modern contraceptive use among the study sample was statistically significantly higher ($p < 0.05$) than that of a national sample of 15- to 19-year-old adolescents (21.4%), as reported by the 2013 national demographic and health survey in the DR [48].

*3.5. In-Depth Interviews*

Results from qualitative analysis of interview transcripts ($n = 27$) indicated participants felt they learned a lot during the intervention and, related to our quantitative measures of social support and autonomous functioning, participants indicated they had received social support in various dimensions and were more capable of autonomous functioning after participation. Overall, participants indicated they were highly satisfied with the intervention and all participants interviewed indicated they would recommend participation in FAMA to others.

### 3.5.1. What Participants Learned

Twenty-four participants (88.9%) discussed having learned new information from their participation in FAMA. When participants were asked to provide specific examples, 78% mentioned learning the family planning topic and emphasized that they learned as much about what does not

work to prevent pregnancy as what does. For example, one participant said, *"about family planning, I loved that. There are some methods that I did not know about and through the group I learned...and that there are some myths that they [people in general] say, supposedly to not get pregnant, but in truth they are not true, like urinating, that does not prevent a pregnancy."* Similarly, another commented, *"bathing after you have sex with your husband, to prevent a pregnancy, knowing that this is false, and I learned, yes, a lot from that, from [family planning] and that this is false."* Yet another said she learned *"to understand things, let's suppose, the myths that people say that after people have had sexual intercourse, that afterwards washing yourself and urinating that that prevents pregnancy and all those are myths because none of that is real."* One participant said of the family planning topic, *" . . . they [the moderators] taught me many things for our babies and for ourselves too, of family planning for ourselves, to not get pregnant at an early age so that we can continue our studies".*

Of the participants, 63% expressed they had learned from and/or valued the topic "understanding your baby" and 55.6% indicated the same for the topic "feeding your baby". One participant comprehensively expressed the importance of both topics: *"I have learned a lot. First, I learned how to care for my baby, I know now when she cries it's not because she is misbehaving, but because something is wrong, or something hurts. I have learned to appreciate her smile, how she grabs me or cuddles. And, in the morning when I wake her up the first thing she looks for the breast, like someone who says: 'Give me breast, give me breast I'm hungry', but she doesn't cry in the morning. The first days it was very difficult to get used to it because she was crying and I did not know why, because I am a first time mother . . . but thank you to FAMA, I have learned a lot about breastfeeding. My mom used to say to me: 'From the moment the child is born, you stick a bottle of milk to them', it is not like that, the child is given the breast for the first 6 months".*

### 3.5.2. Social Support

There was evidence of increased social connectedness, companionship, informational support, and emotional support demonstrated in the interviews. Nearly all participants (96.3%) expressed feelings or perceptions of increased social connectedness and/or the availability of someone with whom to share enjoyable social activities as a result of the FAMA intervention. As one participant commented, *" . . . it [the group] was practically a family, they guided us and showed us a lot of care and appreciation. You've seen how [the moderators] are with us, they are very playful, and I thank you and them because I have learned a lot and I have helped my baby a lot...I had to change environments and look for a job and all that so I don't participate as much in the group now, but still I felt like I was in a family, I felt the warmth of being in a home with other girls."* Another commented, *" . . . some of the girls, we got along well, not a simple friendship 'Ah we talk on WhatsApp and that', no we can already say, 'maybe one day we can go to her house, we are going to hang out, we are going to cook with the children', you know? And we built trust."* Emotional support was mentioned by 37.0% of the participants. As one participant said, *"I felt really good and supported. It even lifted my self-esteem; that was something really nice."* Another participant reported a similar self-esteem boost sharing, *"Yes, I even feel that my self-esteem has risen a little because I am a little antisocial and I have opened up a little, speaking in the conversations . . . I have met many people."* One participant shared that one of the facilitators provided emotional support during a difficult time, *"My experience was very comfortable, I felt very trusting, I could communicate, not only in private but also with the girls, many times...I mean I've had a lot of relationship problems with the baby's father, they [the moderators] have helped me a lot, they advise me, I felt comfortable, we shared a lot, I learned a lot..."* Receipt of informational support was expressed by 55.6% of the participants. Many participants shared the sentiment that, *"[the moderators] are really good, they helped me significantly, always attentive to whatever question, really good treatment."* Several participants provided examples of the informational support provided to them including about family planning, what to do when their baby is crying or constipated, among other information received.

### 3.5.3. Autonomous Functioning

There was evidence of increased autonomous functioning in the exit interviews as participants expressed heightened interest and ability to take care of themselves and their babies. As one mother

said, *"I learned to be more awake, like more agile, and to interact more with other people."* Another said, *"before [the intervention] there were things that I didn't feel that kind of trust [to talk about], not even to talk with my mom. Now I do it, I have opened a little, I've left my comfort zone."* Another said, *"Oh, with my husband an example, I feel more... I mean, I tell him things he can do with the child and I explain to him that if he can, he can give him his little meal . . . before he gave him anise and oregano tea, and I said: 'No, you can't give that,' and he said to me, 'okay,' and I explained why."* Similarly, participants indicated decreased levels of susceptibility. As one mother said, *"I learned that I can have my own opinion and decide for myself."* Another demonstrated high authorship saying, *"I would say in five years I will be working. I would also like to dedicate myself to my family and sometimes hang out with my girlfriends. I would like to be like that in the future."*

### 3.5.4. Overall Satisfaction

Overall, participants expressed general satisfaction with the FAMA intervention. As one participant shared, *"It is a very excellent group, very excellent, it should be extended more in the country because it provides significant help to first time moms."* Another said, *" . . . I learned a lot, it was really useful to me the information they gave us, the help with the babies, because I was totally inexperienced."* Every participant expressed they would recommend the FAMA program to others. As one participant summarized, *"this group was really, really special, truly it was. I am really very happy, very happy, I am. The only thing that bothers me is that now we aren't going to be able to participate because this is the end. I was really, really happy in this group, I would pass the day really happy reviewing messages. I wasn't angry, no, because I would grab the phone and when I grabbed the phone it was for me to smile."* One participant said they would recommend the group because, *" . . . sometimes [girls] get pregnant at an early age and they don't know how to take care of their child, but [in the group] they deal with all topics, there are some girls who know more than others, they can help each other."*

### 3.5.5. Recommendations for Future Improvements

When asked what we could do to improve the intervention, one participant gave content-related feedback, suggesting expanding the baby illnesses topic to include more childhood illnesses. Others voiced other concerns related to the function of the group including the difficulty involved in juggling the group with other responsibilities, determining how to meaningfully contribute to the group, annoyance over long waits for responses to questions, and complaints that some participants did not follow group norms. For example, one participant said, *"Sometimes they talked a lot, I practically didn't talk, I only entered [into the chat] to see what interested me, and I didn't like it, that they talked so much, many voice memos, talked about things that they shouldn't be talking about and that's why I didn't talk much in the group."* Some were frustrated by the lack of participation by some of the participants. One participant said, *"in reality, I didn't like it when other participants didn't talk much in the group . . . that bored me a little. But, I liked it [FAMA] a lot, it was very interesting."*

## 4. Discussion

As we move toward an increasingly digital and global world, it is important to effectively design and evaluate technologies that can provide information and improve health among those living in hard-to-reach areas. In this study, we aimed to determine if participation in a digital educational support group administered through WhatsApp Messenger centered on the specific needs of new adolescent mothers was a feasible and efficacious method to improve their health knowledge and behaviors. Overall, participants demonstrated statistically significant improvements in postpartum and neonatal knowledge, better attendance at well-baby visits, and as a group, had a markedly higher modern contraceptive use than other adolescents in the Dominican Republic after participating in the intervention. Additionally, participants indicated feeling increased social connectedness as well as higher levels of informational and emotional support following participation.

A particular strength of our study is the evaluation component, as many studies detailing the development of digital technologies for limited-resource settings focus on the feasibility,

acceptability, and design of new technologies, while few evaluate the outcomes observed following their interventions [52]. Furthermore, our use of a mixed-methods study provides sound evidence and a rich description regarding adolescents' perspectives of the intervention as well as the improvements in outcomes experienced from their participation in it. Our findings, therefore, provide a valuable contribution to the literature, as evidence-based methods to help adolescent mothers living in limited-resource settings manage the postpartum period are acutely needed and hard to come by. Additionally, evidence-based methods for the effective evaluation of these interventions are needed [38]. Our findings are unique as much of the research on WhatsApp interventions and adolescent pregnancy focuses on sexual and reproductive health, including sexually transmitted infections and pregnancy prevention [53,54]. Our findings were, however, similar to one recent study conducted in Kenya, which found a mobile support group offered through WhatsApp is safe, feasible, and enhanced rates of postpartum contraceptive uptake among participants [27]. That these types of interventions are demonstrating efficacy among similar populations in as diverse of settings as the DR and Kenya further emphasizes that well-designed digital interventions are a promising method to improve health outcomes among adolescents in limited-resource settings globally.

Beyond our findings, our experience implementing our intervention provided many important lessons that are applicable to future digital interventions. First, it took extensive planning and coordination to determine how to provide cell phones to each participant as well as ensure participants had sufficient internet connectivity or cellular data to be able to participate in the intervention. Despite numerous global statistics indicating there is widespread smartphone use in limited-resource settings [55,56], other studies have identified disparities in who has access to phones or that cell phone connectivity is very limited [25,54,57]. Before implementing mHealth or other digital technologies in limited-resource settings, exploring target participants' access to cellular phones and internet connectivity is an important first step. Then, once we provided participants cellular phones, they quickly exhausted their data packages by sending photos of their babies and voice memos in the WhatsApp groups. In response, we purchased weekly data plans that enabled more internet and cellular data use and implemented "group rules" that limited the number of pictures/videos and length of voice notes that could be sent. These are important lessons when considering implementing digital interventions, as a thorough understanding of target participants' potential data use, and how to obtain sufficient connectivity to maintain it, is critical. Additionally, we had the funding to provide phones and data packages to intervention participants from a successful grant application. As this may not be the case for many organizations looking to implement this type of intervention, some possible ways to reduce costs are to include less participants, only include participants who have access to a cellular phone and WhatsApp, as has been done in other studies [27], or place limits on the amount of data they are able to use.

Encouragingly, the WhatsApp groups were very active. However, this meant the group moderators had to dedicate more time than anticipated to answer questions, maintain group norms, and correct misinformation. To help manage participants' expectations (i.e., that moderators would not respond in the middle of the night), a group norm was developed to clarify that although participants could send messages at any time, the moderators would only respond Monday–Friday during working hours. Beyond monitoring the group chats, responding to individual messages received from participants was a time-consuming task for moderators. The large group size may have exacerbated this problem. For future interventions, it will be important to recognize the time and resources required to manage a support group such as this and establish the norms around, expectations for, and availability of moderators. The size of the digital groups should also be carefully considered, as of course larger groups will maximize impact and a sufficient number of participants is needed to generate meaningful conversation as well as social and peer support. However, groups should not be too big that vocal participants drown out their more timid counterparts and moderators cannot reasonably manage questions received from all group members. Another consideration of the time investment by moderators is the associated cost, which may reduce the cost effectiveness of the intervention.

To mitigate this concern, the amount of time moderators are able to spend on the intervention could be more clearly regulated at the outset of the study, participants could be limited on the number of questions they can pose to moderators outside of the group chat, or automated options, such as effectively designed chatbots, could be used to answer frequently asked questions [58]. It is, however, notable that other studies in limited-resource settings have also been able to have moderators involved in WhatsApp groups in a cost-effective way [27], but since WhatsApp groups are still emerging as a health-related intervention method, further analyses on what makes them cost effective are warranted.

Two additional considerations important for future studies are informed consent and privacy concerns with the digital groups. As previously mentioned, in the DR, pregnant adolescents are considered emancipated minors who can legally provide informed consent to participate in research studies. As this is not always the case, future intervention designers must consider how to obtain consent for younger adolescents' participation. Privacy concerns are another large issue that may affect similar interventions in different contexts. Unlike many messaging platforms, WhatsApp Messenger has end-to-end encryption [26], which makes it a more secure platform. However, it may not meet all privacy standards established to protect the sharing of personally identifiable health information, as might be a concern in similar support groups. In this case, knowledge of local privacy laws and regulations is paramount, as is careful attention to the informed consent process so that potential participants understand what they are committing to when they join the group. Furthermore, creating group norms that limit the sharing of private medical information in the group may minimize this concern.

This study had limitations. First, 16 participants had previously participated in the user-centered design sessions conducted during the intervention design stage. Those participants could, therefore, have been more comfortable with the study team and MAMI clinic, more engaged in their health and/or health care, and more willing to participate in groups. Notably, initial design sessions only explored participants' experiences as new mothers and their corresponding information needs; participants in those sessions did not receive any type of patient education or health coaching. Therefore, it is unlikely participation in these sessions led to increased knowledge of postnatal concepts or an increase in contraceptive use. A second limitation was that in-depth interview participants were a convenience sample of mothers who attended the final in-person group sessions, which could lead to bias as those in attendance were likely the more engaged participants. Additionally, despite implementing methods such as indirect questioning [59,60] to avoid it, social desirability bias may have caused participants to overestimate or overstate positive feelings toward the intervention during in-depth interviews. Additionally, retrospective data collection of medical record data of participants and controls led to a large amount of missing vaccination data and therefore, the vaccination data analysis was omitted from the study. Lastly, the contraceptive methods used by paired controls was frequently missing in MAMI medical records. This made it so that we could not compare contraceptive use of participants and controls, which we remedied by substituting a national sample for comparison. This highlights an important concern in the record keeping at this clinic and is an important reminder to those working in similar settings globally; thoroughly documented medical records are critical to enable effective analysis of retrospective patient data. Regardless of limitations, we were able to successfully conduct the intervention with three separate groups and most participants demonstrated increased knowledge and improved health behaviors post-intervention as compared to controls and a national sample of adolescent females.

## 5. Conclusions

Moderated WhatsApp support groups centered on the needs of new adolescent mothers are feasible for health centers to conduct, acceptable by the target population, and can have a positive impact on health knowledge, perceptions of social support, and health seeking behavior for family planning and well-baby visits. Further study and improved documentation are necessary to determine

impacts on childhood vaccination coverage. Our results emphasize that digital technologies are a useful and important technology that can lead to better health outcomes in limited-resource settings globally.

**Author Contributions:** Conceptualization, S.S., M.H., D.B., and L.M.; methodology, S.S., M.H., L.M., E.H., L.G., and E.D.; software, E.H., A.L., S.S., and K.A.P.; formal analysis, E.H., S.S., and A.L.; investigation, V.A., L.M., K.A.P., E.D., and L.G.; validation, S.S., E.H., A.L., L.G., E.D., D.B., K.A.P., V.A., L.M., and M.H.; resources, S.S., and E.H.; data curation, E.H., V.A., and K.A.P.; writing—original draft preparation, S.S.; writing—review and editing, E.H., A.L., L.G., E.D., D.B., and M.H.; supervision, L.M., M.H., and S.S.; project administration, V.A., and K.A.P.; funding acquisition, D.B., M.H., and L.M. All authors have read and agreed to the published version of the manuscript.

**Funding:** This project was supported by Grand Challenges Canada award number ST-POC-1808-17557. Grand Challenges Canada is funded by the Government of Canada and is dedicated to supporting Bold Ideas with Big Impact®. The author SS was funded through National Institute of Nursing Research of the National Institutes of Health under Award Number K99NR017829. The content is solely the responsibility of the authors and does not necessarily represent the official views of the National Institutes of Health. Also, the findings and conclusions in this article are those of the authors and do not necessarily represent the views of Planned Parenthood Federation of America, Inc.

**Acknowledgments:** The authors would like to send a special thank you to our participants who were generous with their time and involvement in this study.

**Conflicts of Interest:** The authors declare no conflict of interest. The funders had no role in the design of the study; in the collection, analyses, or interpretation of data; in the writing of the manuscript, or in the decision to publish the results.

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
