# Peer review of "Digital Educational Support Groups Administered through WhatsApp Messenger Improve Health-Related Knowledge and Health Behaviors of New Adolescent Mothers in the Dominican Republic: A Multi-Method Study"

_informatics, doi:10.3390/informatics7040051_

Round 1
Reviewer 1 Report
This is an interesting and well-written paper regarding use of the WhatsApp messaging platform to support adolescent mothers, during the postpartum period, in the Dominican Republic. It is noted that the authors did a particularly good job of discussing the potential limitations of their study, as well as highlighting some of the challenges related to implementation of digital health initiatives in low/middle-income country settings.
Please address the following comments:
- Section 2.6.2: It is requested that the authors clarify, in the methods section, the 3 groups that were utilized in this study. "Each of three FAMA intervention groups (n=16, n=20, n=22)..." These 3 groups are not previously described; prior to this point in the manuscript, it appears that there are 2 groups in the study -- an educational intervention group (through WhatsApp) and a control group. Please provide further details so that readers have a clear understanding of the study design.
- It is noted that in paragraph 3.5.4, the quotes from the in-person interviews should be put in italics, to match the convention utilized in the other sections.
- Please address the potential issue of confounding, due to the fact that 16 of the subjects also participated in the user-design phase of the study. What was the potential impact (if any), related to the fact that these users may have felt much more engaged/invested within the process, and may, or may not, have been primary drivers for some of the other results?
- There is a reference, in the discussion, to attempts to ascertain impact of the intervention on vaccination rates, but the discussion is the first place this is mentioned. Please be sure to include this in the methods section, as well.
Reviewer 2 Report
I like the paper very much. The authors present positive results of a WhatsApp-based intervention on young adolescent mothers. The paper is well written and provides useful insights for other interventions.
I have a number of minor issues / suggestions:
- Page 2, line 23, you say that WhatsApp is among the most secure messaging apps. I think this is too optimisitic; most apps provide end-to-end encryption; there are more secure and privacy-oriented apps (in the discussion, you provide a bit more nuance, please also do that in the introduction).
- You do not discuss the possible effects of the fact that a number of participants already participated in the initial design session (page 3, line 25). Please add that to the discussion.
- You compare the intervention group with a similar group of patients, however that only becomes clear at page 5 (section 2.8). Please discuss that earlier.
- Also, discuss whether the pre-intervention meeting could have had a positive influence on the study group (ideally, the control group would also have had a similar meeting).
- Page 7, section 3.4.2: this was a pity. I suggest that you make more explicit that the control group for this measure did not necessarily had a baby. This makes the conclusions for this measure less strong.
- Page 10, discussion: you describe the large effort that moderators had to put in the intervention; could you say something about the cost-effectiveness of your intervention compared with other interventions (by comparing with literature)? If not, include this point in the discussion as open question.
- You could also discuss whether there is potential in partly automating the responses; e.g. do you think that some often occuring questions can be answered by chatbots?
